# Central Nervous System Tuberculosis in a Murine Model: Neurotropic Strains or a New Pathway of Infection?

**DOI:** 10.3390/pathogens13010037

**Published:** 2023-12-30

**Authors:** Daniel Rembao-Bojórquez, Carlos Sánchez-Garibay, Citlaltepetl Salinas-Lara, Brenda Marquina-Castillo, Adriana Letechipía-Salcedo, Omar Jorge Castillón-Benavides, Sonia Galván-Arzate, Marcos Gómez-López, Luis Antonio Jiménez-Zamudio, Luis O. Soto-Rojas, Martha Lilia Tena-Suck, Porfirio Nava, Omar Eduardo Fernández-Vargas, Adrian Coria-Medrano, Rogelio Hernández-Pando

**Affiliations:** 1Departamento de Neuropatología, Instituto Nacional de Neurología y Neurocirugía Manuel Velasco Suárez, Insurgentes Sur 3877, Tlalpan, Ciudad de México CP 14269, Mexico; jdrbojorquez2002@gmail.com (D.R.-B.); carlos.s.garibay@live.com.mx (C.S.-G.); mltenasuck@gmail.com (M.L.T.-S.); 2Programa de Doctorado en Ciencias Quimicobiológicas, Escuela Nacional de Ciencias Biológicas, Instituto Politécnico Nacional, Prolongación de Carpio y Plan de Ayala s/n, Col. Santo Tomás, Ciudad de México C.P. 11340, Mexico; lajimenezz@gmail.com; 3Red MEDICI, Carrera Médico Cirujano, Facultad de Estudios Superiores Iztacala, Universidad Nacional Autónoma de México, Tlalnepantla 54090, Mexico; oskarsoto123@unam.mx; 4Tuberculosis Research Commonwealth, Mexico City 14269, Mexico; 5Programa de Doctorado en Ciencias en Investigación en Medicina, Escuela Superior de Medicina, Instituto Politécnico Nacional, Plan de San Luis y Díaz Mirón s/n, Col. Casco de Santo Tomas, Alcaldía Miguel Hidalgo, Ciudad de México C.P. 11340, Mexico; 6Laboratorio de Patogénesis Molecular, Laboratorio 4 Edificio A4, Carrera Médico Cirujano, Facultad de Estudios Superiores Iztacala, Universidad Nacional Autónoma de México, Tlalnepantla 54090, Mexico; 7Departamento de Patología, Instituto de Ciencias Médicas y Nutrición Salvador Zubirán, Vasco de Quiroga 15, Belisario Domínguez Secc 16, Tlalpan, Ciudad de México 14080, Mexico; brenda.marquina@yahoo.com.mx; 8Laboratorio Clínico, Instituto Nacional de Neurología y Neurocirugía Manuel Velasco Suárez, Insurgentes Sur 3877, Tlalpan, Ciudad de México CP 14269, Mexico; audra_fnx@hotmail.com; 9Centro Neurológico del Centro Médico ABC, Av. Carlos Fernández Graef 154, Santa Fe, Contadero, Cuajimalpa de Morelos, Ciudad de México 05330, Mexico; castillon_omarjorge@hotmail.com; 10Laboratorio de Neuroquímica, Instituto Nacional de Neurología y Neurocirugía Manuel Velasco Suárez, Insurgentes Sur 3877, Tlalpan, Ciudad de México CP 14269, Mexico; sonia_galvan@yahoo.com; 11Instituto Nacional de Rehabilitación (INR) “Luis Guillermo Ibarra Ibarra”, México City 14389, Mexico; golma7621@hotmail.com; 12Departamento de Fisiología, Biofísica y Neurociencias, Centro de Investigación y de Estudios Avanzados del Instituto Politécnico Nacional, Mexico City 07360, Mexico; pnava@fisio.cinvestav.mx; 13Servicio de Hematología del Instituto Nacional de Cancerología, Av. San Fernando 22, Belisario Domínguez Secc 16, Tlalpan, Ciudad de México 14080, Mexico; ofernandezvargas@icloud.com; 14Programa de Maestría en Ciencias en Neurobiología, Instituto de Neurobiología, Campus UNAM-Juriquilla, Juriquilla, Querétaro 76230, Mexico; coria.medrano.adrian@gmail.com; 15Sección de Patología Experimental, Instituto de Ciencias Médicas y Nutrición Salvador Zubirán, Vasco de Quiroga 15, Belisario Domínguez Secc 16, Tlalpan, Ciudad de México 14080, Mexico

**Keywords:** central nervous system tuberculosis, experimental tuberculosis, neurotropism, tuberculosis animal model

## Abstract

Tuberculosis (TB) of the central nervous system (CNS) is a lethal and incapacitating disease. Several studies have been performed to understand the mechanism of bacterial arrival to CNS, however, it remains unclear. Although the interaction of the host, the pathogen, and the environment trigger the course of the disease, in TB the characteristics of these factors seem to be more relevant in the genesis of the clinical features of each patient. We previously tested three mycobacterial clinical isolates with distinctive genotypes obtained from the cerebrospinal fluid of patients with meningeal TB and showed that these strains disseminated extensively to the brain after intratracheal inoculation and pulmonary infection in BALB/c mice. In this present study, BALB/c mice were infected through the intranasal route. One of these strains reaches the olfactory bulb at the early stage of the infection and infects the brain before the lungs, but the histological study of the nasal mucosa did not show any alteration. This observation suggests that some mycobacteria strains can arrive directly at the brain, apparently toward the olfactory nerve after infecting the nasal mucosa, and guides us to study in more detail during mycobacteria infection the nasal mucosa, the associated connective tissue, and nervous structures of the cribriform plate, which connect the nasal cavity with the olfactory bulb.

## 1. Introduction

Tuberculosis (TB) is a transmissible, chronic, and granulomatous disease caused by the bacterium *mycobacterium tuberculosis* (Mtb). Primarily, Mtb affects the lungs, although it can attack any organ or body tissue and can present a wide range of clinical manifestations [1]. Central nervous system (CNS) involvement following Mtb infection is one of the most devastating clinical manifestations of TB, with high levels of mortality and incapacitating neurologic sequelae in most survivors [2].

CNS-TB is considered a postprimary form of TB, caused by a spreading of Mtb from the pulmonary parenchyma caused by spreading of Mtb from the pulmonary parenchyma to the CNS. This process has been attributed to lympho-hematogenous dissemination after the rupture of Ghon’s foci, in which free mycobacteria colonize the vascular endothelium of arachnoid mater vessels and cause structural damage to the blood–brain barrier (BBB) [3,4,5] and by infected macrophages that cross paracellular vascular endothelium to CNS, “The Trojan horse mechanism” [6].

Furthermore, evidence suggests that a pulmonary infection is not always necessarily the primary organ of infection, and that certain Mtb genotypes have been described that could have a predilection to colonize and invade the CNS in the early stages of infection, coining the terms neurovirulent and neurotropic strains [7,8].

Indeed, our group reported in BALB/c mice infected by the intratracheal route, there was a rapid CNS dissemination of Mtb strains with identical *spoligotypes* from Colombian patients with tuberculous meningitis [9]. Therefore, these results raise the question of whether pulmonary infection is necessary before CNS infection and the existence of neurotrophic bacteria. In this work, the same Mtb strains were used to infect BALB/c mice by the intranasal route, we detected viable Mtb in CNS tissue before the microbiology or histologic evidence of pulmonary disease, suggesting a novel alternate route to colonization/invasion of the CNS by specific Mtb strains.

## 2. Materials and Methods

### 2.1. Ethical Approval

All animal studies were conducted in a bio-safety level III facility, in accordance with the Mexican constitution statute NOM 062–Z00-1999. The experiments were performed with the consent of the INCMNSZ’s Ethical Committee for Animal Experimentation, under protocol number CINVA: 223.

### 2.2. Mycobacterial Strains

*M. tuberculosis* strains 28, 136, 209 from the Latin American Mediterranean (LAM) genotype family with the st. 33, spoligotype 776177607760771, all isolated from CNS TB patients, and the reference strain H37Rv were obtained as previously described [9]. Briefly, 0.5 mL of mycobacterial suspension was used to inoculate 8 mL of Middlebrook 7H9 Broth (Difco Lab, Detroit, MI, USA) supplemented with Oleic Albumin Dextrose Catalase (OADC) (Difco Lab, Detroit, MI, USA), and incubated at 35 °C with constant agitation for 72 h. At the end of this period, the culture was scaled up by transferring 5.0 mL of the bacterial growth broth to 60 mL of sterile BD^®^ 7H9 broth maintained at a temperature of 35 °C. Bacterial growth was indirectly measured through the optical density (OD) of the culture at 600 nm, and the culture kinetics were monitored to generate growth curves for each strain. Parallel cultures were performed under the same conditions to verify purity through blood agar plate streaking and Ziehl–Neelsen staining.

All the Mtb strains were collected during their early or mid-logarithmic growth phase (approximately at an OD600 nm = 0.600 for this strain), which was determined based on growth curves. To harvest Mtb for mice infection, bacteria were transferred to 50 mL conical tubes and centrifuged at 3000 rpm at 4 °C for 5 min, and the supernatant was discarded. Subsequently, an approximate volume of 5 mL of sterile glass beads was added, and the volume was brought up to 15 mL with 1X PBS-0.05% Tween. To disaggregate the bacterial pellet, 5 cycles of vortex agitation were performed, consisting of 1 min of agitation and 1 min of rest. The supernatant was collected in a new 50 mL conical tube, and 1X PBS was added to reach a volume of 45 mL. They were centrifuged at 2500 rpm at 4 °C for 5 min, and the supernatant was discarded using a pipette to preserve the integrity of the cell pellet. Finally, 15 mL of filtered sterile physiological saline solution (FSPSS) was added, the supernatant was removed with a pipette, and the pellet was resuspended in 10 mL of FSPSS.

Inoculum was adjusted to 250,000 bacteria in 46 μL, and a viable count was performed using flow cytometry with the Guava Easy Cyte Mini System, following the technique described by the manufacturer. The technique is based on labeling all mycobacterial cells with the fluorochrome FITC and labeling dead cells with IP, a nucleic acid base analog. Scatter plots are generated and analyzed with FlowJo software v.10 (FlowJo v.10, Ashland, OR, USA), which calculates the concentration of mycobacteria in the sample. An appropriate dilution is then made to obtain 3 mL of the inoculum at the previously mentioned concentration. With the adjusted inoculum, serial dilutions were performed to verify the accuracy of the equipment through viable count by agar plate streaking on 7H10 agar. The bacterial inoculum of the different strains was kept on ice during transportation to the animal facility and throughout the entire infection process.

### 2.3. Experimental Murine Model

In total, 24 male BALB/c mice 2–3 weeks old per group were inoculated with Mtb strains. Mice were anesthetized with sevoflurane in an acrylic chamber of 8 × 6 × 6 inches saturated with vapors of 10 μL of sevoflurane as anesthetic [9]. After this, the mouse was placed in a supine position and slowly inoculated intranasally with 2.5 × 10^5^ bacilli of the corresponding strain. Afterwards, they were sacrificed by exsanguination at days 1, 3, 7, 14, 21, 28, 60, and 120. Mice were anesthetized with intraperitoneal Penthotal. Once they lost consciousness, they were placed on a dissection table, secured by all four limbs and the head. Asepsis and antisepsis were performed using 96% (*v*/*v*) ethanol. Animals under Pentothal anesthesia were euthanized by cutting the left axillary artery. Encephalon, liver, lung, kidney, and spleen were removed. The obtained tissues were immersed in 4% formaldehyde at a 1:10 volume-to-volume ratio and allowed to fix for 48 h to preserve their antigenicity. Encephalon was further dissected in the olfactory bulb, brainstem, and brain hemispheres. In total, 3 whole heads were reserved to evaluate encephalon and nasal cavity structures in a full morphofunctional context. A survival study was performed for 16 weeks. In total, 20 mice from each group were left undisturbed to the record. Overall, 3 separate experiments were performed. 

### 2.4. Histology Sample Preparation

Tissues were dehydrated and paraffin-embedded using a Histokinett, following a 12 h cycle, with increasing concentrations of alcohols, xylenes, and paraffin. Formalin-fixed tissues were embedded in paraffin, sectioned at 2 μm thickness, and stained with either hematoxylin and eosin (H&E) or Ziehl–Neelsen (ZN) stain. Whole heads were decalcified in ethylenediaminetetraacetic acid (EDTA) 0.5 M pH 8 for 3 weeks at 5 °C, after formalin fixation. The tissues underwent processing using the conventional histological technique for hematoxylin and eosin (H&E) staining. Subsequently, the histological sections were examined under a bright-field binocular microscope.

### 2.5. Determination of Colony Forming Units

Spleen, liver, lung, brain, olfactory bulb, and brainstem or whole encephalon from the mice were placed in 2000 μL conical bottom vials for automated oscillatory homogenization, which contained 125 μL of 0.1 mm glass beads and 125 μL of 0.5 mm beads. To these tubes, 500 μL of phosphate-buffered saline (PBS) solution with a pH of 7.6 was added, and they underwent two cycles of oscillation using a Mini Bead-Beater-1 by Biospect at a speed of 2500 oscillations per minute for 10 s. The tube containing the sample was immediately placed on ice after tissue homogenization.

Homogenized tissues were used to perform microbial load quantification by viable count determination, duplicate decimal dilutions were performed in ELISA plates for each organ using a final volume of 300 μL, homogenizing 10 times. Subsequently, 20 μL of each dilution was taken and streaked by plating on BD^®^ 7H10 medium enriched with BD^®^ ADC, divided into 8 sectors. They were incubated within plastic bags in a 37 °C CO_2_ incubator with a 5% CO_2_ atmosphere and incubated for 14 and 21 days to count colony forming units (CFU). 

### 2.6. Statistical Analysis

Survival curves were analyzed with Kaplan–Meier plots and Long Rank tests. Two-way ANOVA with Tukey post-test was used to determine the statistical significance of bacterial load. *p* < 0.05 was considered significant.

## 3. Results

### 3.1. Superior Lethality, Virulence and Pulmonary Diseminationto of Neurotropic Strains

Mycobacterial strains exhibited varying levels of virulence. As depicted in Figure 1A, infected mice with the different Mtb clinical isolates and reference strain H37Rv as control showed diverse mortality rates: Infection with strain 28 permitted 100% survival, comparable to the control strain H37Rv, whereas strains 136 and 209 exhibited lethality rates of 1.53% and 7.69%, respectively. The only significant difference (*p* < 0.01) observed were between strain 136 and the control strain.

In the evaluation of pulmonary bacterial load, we found that strains exhibited significantly lower infection to the lung, as demonstrated by viable CFU counts compared to the control strain H37Rv throughout almost all the evaluated time points, *p* < 0.0001, except for days 14 and 120, where strain 136 presented a higher load (*p* < 0.0001 in both times and for the comparison for all strains) (Figure 1B).

Lung histological analysis revealed the presence of congestive blood vessels and few peribronchial lymphocytes starting at day 7 in the animals infected with strain 209; from day 14 and onwards, granulomas and pneumonia were detected while the severity of these lesions worsened at later times, until the last day of sacrifice. Representative microphotographs of this clinical strain are presented in Figure 2. Whereas mice infected with the strains 28 or 136 showed similar histopathological alterations at the same time points as those observed in mice infected with strain 209. Meanwhile, in mice infected with strain H37Rv, there were no abnormalities until day 14, when congestive blood vessels and peribronchiolar inflammatory cells were observed. From day 21, granulomas and pneumonic foci were observed, and as the days post-infection progressed, the percentage of areas affected by pneumonia increased. Representative micrographs of the control strain are presented in Figure 3.

### 3.2. Encephalon Histology and Bacillary Load

Encephalon collected from mice infected with strains 28, 136, and 209 showed early infection starting at day 1 and for all strains at day 3 (Figure 4C). Interestingly, strain 209 infected faster with an important increase at 7 dpi (*p* < 0.0001), and their peak detected at day 14 (data results, *p* < 0.0001) followed by a progressive decrease. In contrast, mice infected with strain 136 peaked the CFU in the brain until day 60 (data results, *p* < 0.0001) and remained with *p* < 0.0038 until day 120. Animals infected with strain 28 only showed brain infection on days 21 (data results, *p* < 0.008 with the strain 136 an h37Rv. No CNS invasion was detected with strain H37Rv until day 120 (data results, *p* < 0.0038). Considering that CNS infection by strain 28 was very low, we decided to use strains 136, 209, and H37Rv from this moment on.

We next analyzed the presence of Mtb strains in different structures of the encephalon: olfactory bulb and brainstem. As shown in Figure 4, CFUs in mice infected with strains 209 and 136 were detected in the olfactory bulb and brainstem, starting at day 7. In mice infected with strain 209, CFUs were negative at day 28 in the brainstem, and at day 120 from the olfactory bulb. Meanwhile, Mtb growth in mice infected with strain 136 was negative in the olfactory bulb and brainstem at day 120, however, the infection remained during the whole course of the experiment at the olfactory bulb (Figure 4). It is important to highlight that H&E and ZN staining of the nasal cavity structures, olfactory bulb, cerebellum, and brainstem, of animals infected with all the strains failed to detect either histological abnormalities or acid-fast bacilli, respectively, at the evaluated times. Representative microphotographs of strain 209 are presented in Figure 5 and Figure 6.

### 3.3. Spleen, Liver, and Kidney Mycobacterial Loads

To determine the hematogenous dissemination of Mtb, bacterial loads in different organs were evaluated by CFU quantification

Kidney’s CFUs for mice infected with strain 136 or 209 were detected at day 14 and 21, respectively, and were present till day 120 (Figure 7A). In liver samples from day 14 through 60, we only observed colonization by strain 136, as strains 209 and 136 were undetectable before day 120 (Figure 7B). Mice infected with strain H37Rv did not infect the liver or kidney in these experiments. Furthermore, the bacterial load in the spleen increased from day 14 and was similar between strains 209, 136, and H37Rv until day 28; however, from day 60 onwards, strain H37Rv showed a greater load than the other strains (Figure 7C).

## 4. Discussion

“Dogma dictates that CNS-TB arises after pulmonary lymphohematogenous spread.” This allows the formation of Rich’s foci which, when ruptured, permits mycobacterial dissemination. This model implies that Mtb infection begins elsewhere in the body, mainly in the lung, and then disseminates through the bloodstream to the meninges, where small nodules with bacilli are formed. Nonetheless, the precise mechanisms regulating CNS infection are unknown. Understanding the events involved in mycobacterial CNS infection could facilitate the development of new therapies and profilaxis for CNS-TB. Clinical and epidemiological studies have identified several risk factors for CNS-TB, including host genetic background, but also genetic variations in mycobacterial strains might be involved in their neurovirulence [7,8,10] 

Experimental animal models are a useful tool for studying the immunopathology and pathophysiology of CNS TB. They represent an approach to tracing the histological changes during Mtb infection and have provided the opportunity to elucidate the influences of genetic bacterial diversity in strain virulence and the host immune response [11,12]. In this line of thought, we previously demonstrated rapid CNS dissemination after pulmonary infection with *M. tuberculosis* strains with identical *spoligotype* from Colombian patients with tuberculous meningitis, but not in mice infected with the reference strain H37Rv using a murine model of intratracheal infection [9]. Therefore, these results suggested that pulmonary infection is necessary prior to CNS infection by apparently neurotrophic bacteria, but another non-considered infection route as intranasal could also participate in brain infection. Thus, in this work it is demonstrated that this is in fact an alternative route of brain infection that involves the olfactory nerve and olfactory bulb.

In this current work, we reported early CNS invasion from day 1 post-infection after intranasal bacteria inoculation, with low bacillary load in other organs. Furthermore, although we observed detectable mycobacteria in the lungs of mice infected with these apparently neurotropic strains at early times, histopathological abnormalities following the classical model of mycobacterial CNS invasion, i.e., Gohn’s foci, were missing at those time points. Thus, these results suggested that strains 28, 136, and 209 could invade the CNS using a mechanism different from the classical lymphohematogenous spread. In that respect, we detected mycobacterial loads in the olfactory bulb during the whole course of the experiment, which suggests that the process of CNS infection may start at either peripheral nerves or lymphatic vessels from the nasal mucosa. This would allow Mtb to reach the olfactory bulb and establish CNS infection. However, at the evaluated times in the analyzed tissue, we did not observe histological alterations in the nasal mucosa of infected mice. In a similar way to mycobacterium leprae, it seems that this bacterium is able to colonize and invade the human nasal mucosa without inducing anatomopathological changes [13,14,15]. Regarding the bacterial mechanism for CNS invasion by some Mtb with apparent neurotropism, many molecules could participate, such as the histone-like protein (HLPMt) from Mtb that displays high identity with *mycobacterium leprae* laminin-binding histone-like protein (ML-LBP), which is a molecule that has been implicated in the infection of Schwann cells by *M. leprae* [16,17,18,19]. Another mycobacterial molecules that could be involved are the phenolic glycolipid (PGL-1), this lipid also allows the adhesion of the mycobacterial strains to Schwann cells. The heparin-binding hemagglutinin (HBHA) is a protein related to Mtb dissemination by altering actin polymerization, this protein enables the colonization of pneumocytes and the posterior transcellular migration [20,21] and the mycobacterial kinase pknD, which is important in CNS infection through the adhesion to endothelial cells of the cerebral microvasculature, but is not necessary for pulmonary disease [22,23].

In patients with leprosy, regulatory T cells (Tregs) are more abundant in the lepromatous form, suggesting a pathogenic role in multibacillary presentations. Tregs in paucibacillary forms produce more IL-17 and less IL-10, promoting inflammation and tissue damage. They also secrete IL-35, inhibiting proinflammatory cytokines. The increase in IL-35 correlates with a bacterial load in leprosy, being higher in multibacillary forms. Furthermore, the study found that Tregs in patients with multibacillary leprosy express high levels of PD1 and PDL-1, implying their ability to induce effector cell apoptosis and prevent self-apoptosis. In contrast, in paucibacillary leprosy patients, Tregs express elevated levels of CD95L, possibly related to apoptosis pathways that reduce Tregs. These findings indicate distinct suppression mechanisms and inefficient inflammation regulation, potentially contributing to the persistence of multibacillary leprosy during *M. leprae* infection [24,25].

In this work, we tried to emulate the natural way of CNS infection by Mtb which is the respiratory route. Other authors have injected Mtb directly in the brain, i.e., [26], and report that the inflammatory infiltrate, activation of microglia, and bacterial growth in LCR cultures in BALB/c and DBA/2 mice infected with *M. bovis* BCG. Likewise, Ref. [27] described a murine model with intracranial Mtb infection that developed cerebral tuberculomas and meningitis. To better simulate the natural infection way, we used a model of the nasal mucosal route of infection to investigate the invasion and spread of Mtb to CNS, which demonstrate early infection of the olfactory bulb followed by brain infection. During late infection, when the lungs are affected, it is also possible that Mtb could reach the brain through the bloodstream. In this respect, it has been shown that pknD is required for Mtb to infect brain endothelial cells, but not macrophages or lung epithelial cells. Thus, the Mtb invasion of blood–brain barrier cells could disrupt intercellular adhesions and allow CNS infection, as previously reported [28,29], who infected female Balb/c mice by the intravenous route, detecting CFUs in all regions of the encephalon, and as found in our study, there were no histological changes that were consistent with low levels of inflammatory cytokines in the brain, in contrast to those expressed in lungs that were very high. This anti-inflammatory environment is due to the presence of Treg cells in cerebral tissue. Therefore, the distinctive neurovirulence observed experimentally in different *M. tuberculosis* strains could be due to complex interactions of the host and bacterial genetic background and the environment [30,31]. This could be extrapolated to some clinical cases of cerebral TB in which patients do not exhibit pulmonary lesions or evidence of infection in other organs, suggesting direct arrival of Mtb to CNS that as suggested by our results could be toward nervous structures of the nasal cavity, without leaving aside the possibility that has not been yet confirmed so far of a resolved pulmonary Mtb infection that produced latent Mtb in the lung and brain, being only the latter that suffered reactivation (Figure 8) [32,33,34]. In this regard, in a non-published study carried out with 105 patients with CNS TB at the National Institute of Neurology and Neurosurgery in Mexico City, we found evidence of pulmonary TB in only 22.6% of the patients, which frequently exhibit progressive neurological deficits, with or without non-specific febrile syndrome, and no evidence of imaging or Mtb culture, or other microorganisms. Thus, it seems that other unexplored or unrecognized pathways of infection should participate besides the bacterial dissemination by hematogenous route, in a similar way that the clinical isolates that we used in the present study.

In conclusion, the apparent neurotropic strains that we studied in this work not only exhibit fast lymphohematogenous dissemination but can also reach the CNS by bypassing the lung, infecting first the olfactory nasal mucosa and olfactory nerve. However, additional investigations in other animal models are necessary to confirm that certain Mtb strains can actually perform this. This might involve using Schwann cells that coat the axons of the olfactory neurons as a target to colonize and follow the tract of the septal olfactory nerve across the cribriform plate, providing a direct pathway to the CNS. Such a pathway should be confirmed through more precise and in-depth histological studies, particularly by careful analysis of the nasal mucosa at earlier times after infection.

## Figures and Tables

**Figure 1 pathogens-13-00037-f001:**
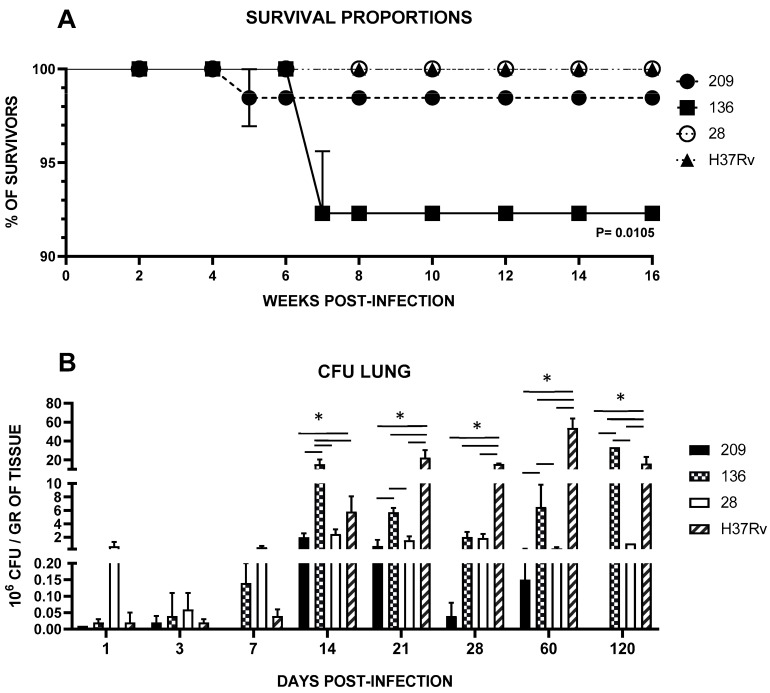
(**A**) Survival Kaplan–Meier plot; (**B**) Lung bacillary load survival curves were constructed with 20 infected mice per bacterial strain and analyzed with Kaplan–Meier plots and the Log Rank test (Mantel-Cox). Data are the mean ± SD of measurements from three mice per time point in three different experiments. Asterisks represent statistical significance (*p* < 0.05) when compared with H37Rv strain infection.

**Figure 2 pathogens-13-00037-f002:**
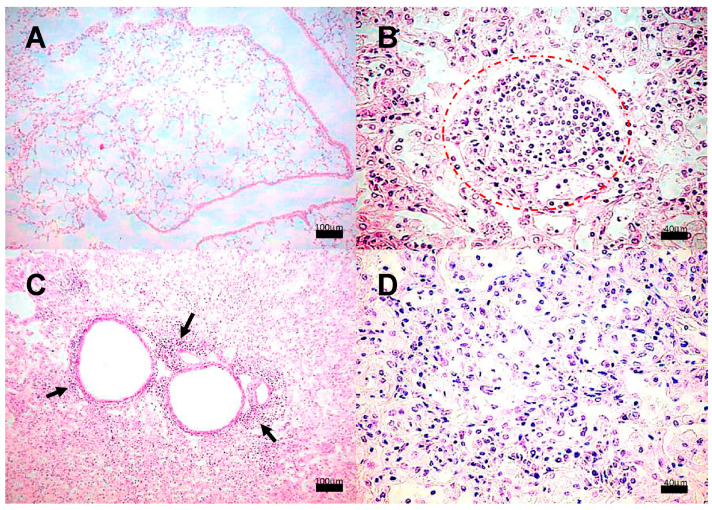
Representative BALB/c mice lung histopathology after infection with *M. tuberculosis* strain 209. (**A**) Tissue without histopathological changes at 1 dpi. (**B**) Granuloma, delimited by the dashed line, surrounded by chronic inflammatory infiltrate at 14 dpi, (**C**) Focal pneumonia and chronic inflammatory infiltrate around blood vessels (arrow) and airways at 21 dpi and minimal intralveolar inflammation. (**D**) Extensive pneumonia at 60 dpi. Micrographs (**A**,**C**) are at 100× magnification stained with H&E, and micrographs (**B**,**D**) are at 400× magnification stained with H&E.

**Figure 3 pathogens-13-00037-f003:**
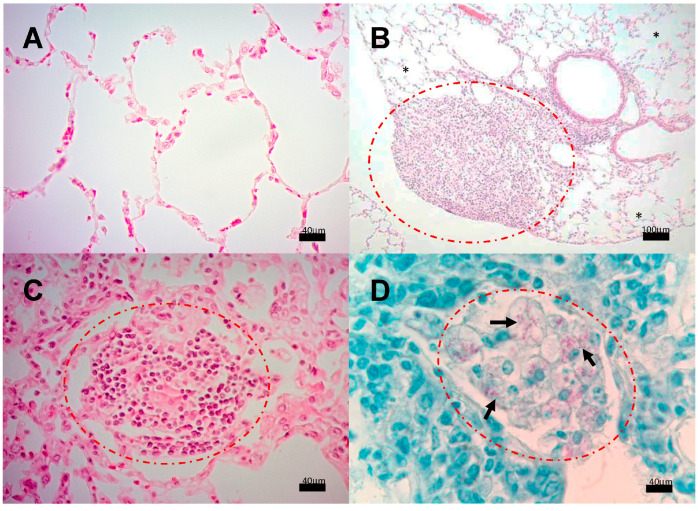
Representative BALB/c mice lung histopathology after infection with *M. tuberculosis* H37Rv. (**A**) Tissue without histopathological changes at 1 dpi. (**B**) Well-consolidated subpleural granuloma, the rest of parenchyma without pneumonia (*) at 14 dpi. (**C**) Granuloma surrounded by an inflammatory infiltrate at 120 dpi, (**D**) Parenchyma with pneumonia, consolidation and acid fast bacilli (AFB) aggregate dyed in fuchsia color (arrows) at 120 dpi, (Micrographs (**A**,**C**) at 400 magnification, micrograph (**B**) at 100 magnifications, stained with H&E, micrograph (**D**) at 400 magnification and stained with ZN, granulomas delimited by the dashed line).

**Figure 4 pathogens-13-00037-f004:**
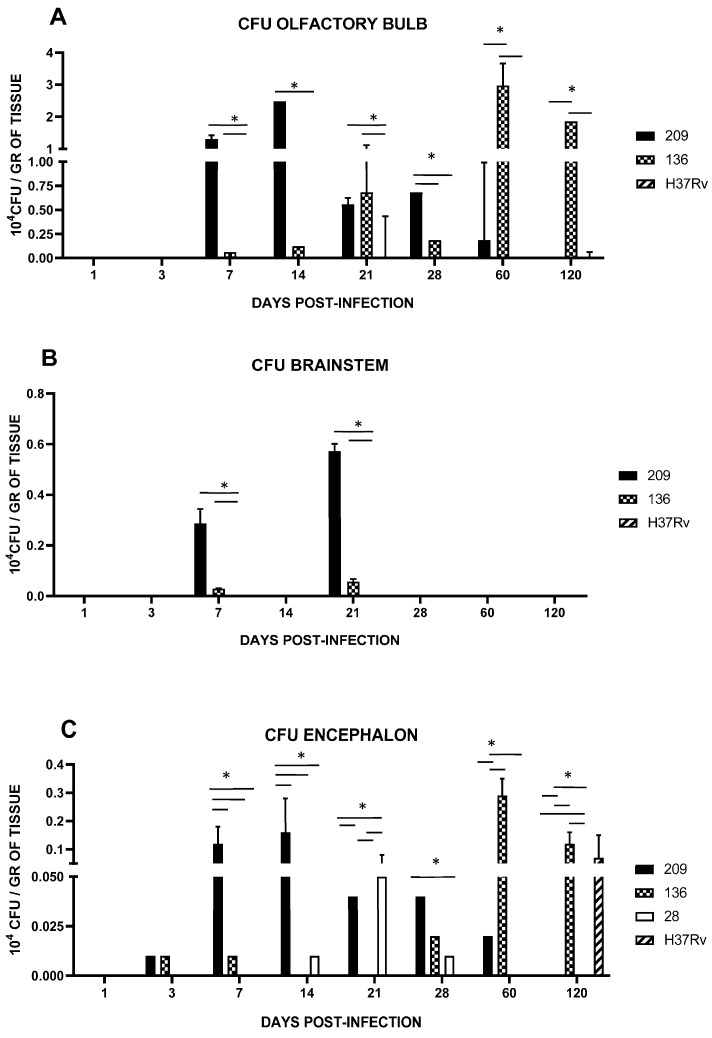
Bacillary load in BALB/c mice infected by an intranasal injection of *M. tuberculosis* isolates of the olfactory bulb (**A**) and brainstem (**B**). Reference strain H37Rv was used as a control. Data are the mean ± SD of measurements from three mice per time point in three different experiments. (**C**) Encephalon bacillary load in BALB/c mice infected by intranasal injection with *M. tuberculosis* isolates. Reference strain H37Rv was used as a control. Asterisks represent statistical significance (*p* < 0.05) when compared with H37Rv strain infection.

**Figure 5 pathogens-13-00037-f005:**
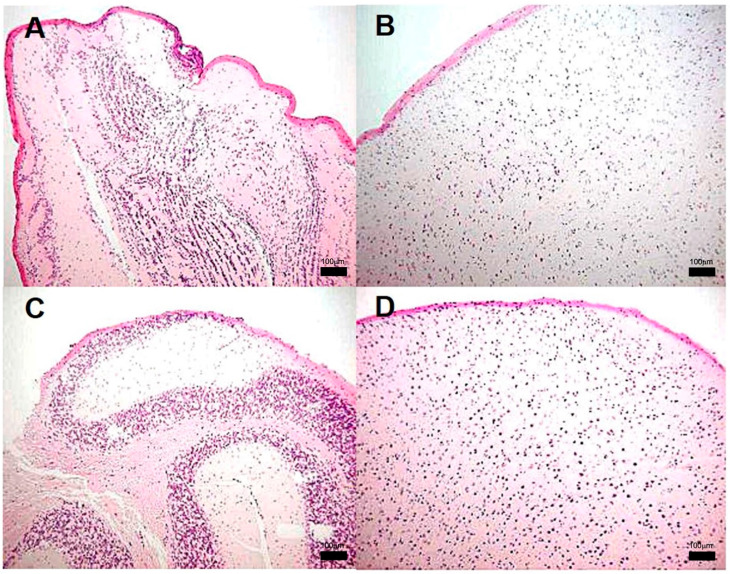
Representative BALB/c mice encephalon histology after infection with *M. tuberculosis* strain 209. (**A**) Olfactory bulb, (**B**) Brain (**C**) Cerebellum, and (**D**) Brainstem. Abnormal Morphological changes were not detected in any section of these anatomic regions. (All micrographs at 100× magnification and stained with H&E at 120 dpi.

**Figure 6 pathogens-13-00037-f006:**
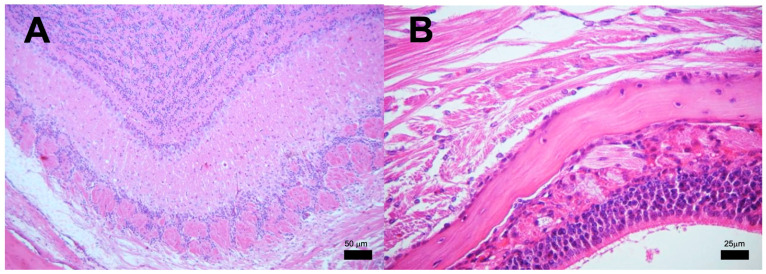
Representative BALB/c mice olfactory area histology after infection with *M. tuberculosis* strain 209. (**A**) Detailed view of the olfactory bulb showing intact tissue without inflammatory or infectious alterations. (**B**) Close examination of the olfactory mucosa, ethmoid bone, and olfactory bulb, none of the structures show histological damage or inflammation that indicate an infectious process. (Micrographs At 100× magnification stained with H&E, micrograph B at 400× magnification stained with H&E at 120 dpi.

**Figure 7 pathogens-13-00037-f007:**
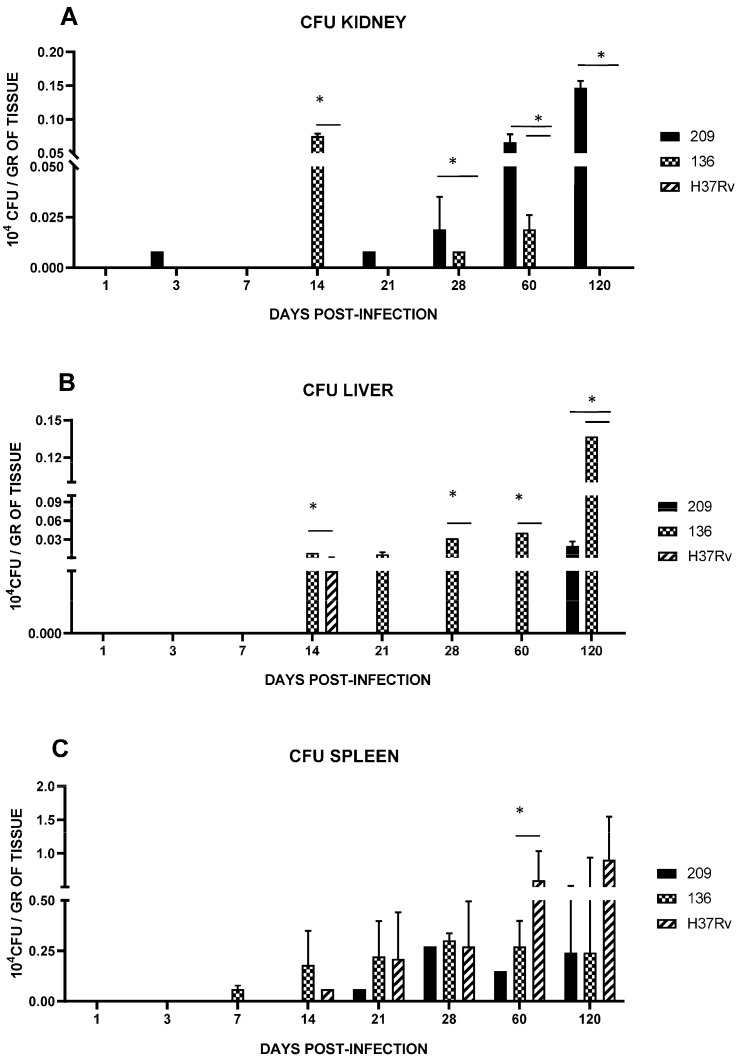
Bacillary load in BALB/c mice infected by an intranasal injection of *M. tuberculosis* isolates of the kidney (**A**), liver (**B**), and spleen (**C**). Reference strain H37Rv was used as control. Data are the mean SD of determinations from three mice per time point in three different experiments. Asterisks represent statistical significance (*p* < 0.05).

**Figure 8 pathogens-13-00037-f008:**
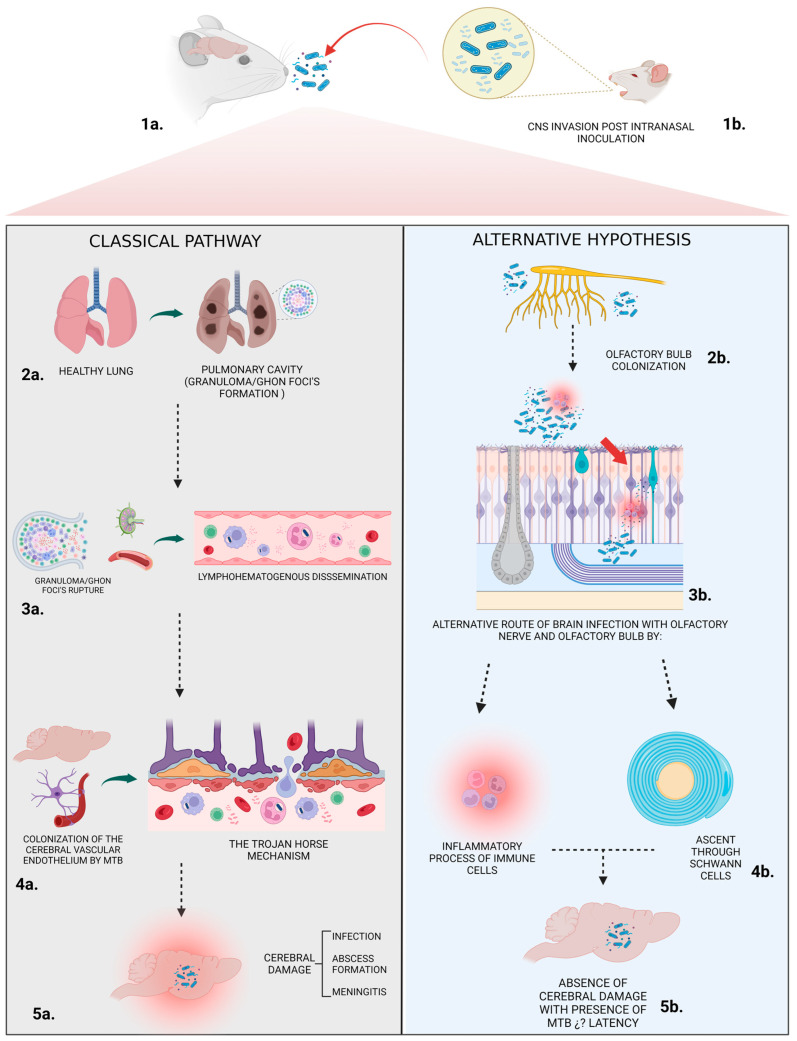
Classical Pathway. Stage **1a**. Initial Infection: Mice inhale droplets containing *mycobacterium tuberculosis* (Mtb). Stage **2a**. Pulmonary Manifestation: In the lungs, Mtb gets engulfed by alveolar macrophages. Within these cells, the bacterium can multiply or remain dormant. The body’s immune response tries to contain the infection, leading to the formation of Ghon’s foci, which are small nodules or lesions. Stage **3a**. Lymphohematogenous Dissemination: If Ghon’s foci rupture, free mycobacteria can enter the bloodstream. Stage **4a**. “The Trojan Horse Mechanism”: This spread causes Mtb to colonize the vascular endothelium of arachnoid mater vessels, leading to damage in the brain–blood barrier (BBB). Infected macrophages (carrying the bacteria) can cross the vascular endothelium into the CNS. This mechanism is likened to the famous “Trojan horse” myth because the bacteria, hidden inside the macrophages, gain entry to otherwise protected sites. Stage **5a**. Cerebral infection. Alternative hypothesis. Stage **1b**. Early CNS Invasion Post Intranasal Inoculation: After introducing Mtb through the nasal pathway, early CNS invasion can be observed from day 1. Stage **2b**. Certain strains of Mtb might not require pulmonary infection to establish a CNS infection. These strains could potentially bypass the lungs and directly colonize the CNS. While mycobacteria are detectable in the lungs, classical signs of CNS invasion, such as Ghon’s foci, are absent at early stages. Stage **3b**. Olfactory Bulb Colonization: Mtb is found in the olfactory bulb throughout the course of infection. This suggests the possibility of CNS infection initiating from peripheral nerves or lymphatic vessels in the nasal mucosa, allowing Mtb to reach the olfactory bulb and establish an infection. Stage **4b**. Mtb’s progression unfolds via two distinct mechanisms: firstly, through the inflammatory response, where activated immune cells strive to contain the pathogen; and secondly, via neural infiltration, where Schwann cells facilitate the bacterium’s ascendancy, echoing the intricate cellular interplay in infection dynamics. Stage **5b**. Absence of cerebral damage: No significant histological changes are observed in the brain of infected mice, homologous to *mycobacterium leprae* behavior in the human nasal mucosa.

## Data Availability

Data is contained within the article.

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
