# Peer review of "Central Nervous System Tuberculosis in a Murine Model: Neurotropic Strains or a New Pathway of Infection?"

_pathogens, 2023, doi:10.3390/pathogens13010037_

Round 1
Reviewer 1 Report
Comments and Suggestions for Authors
The authors propose a novel, alternative route for Mycobacterium tuberculosis clinical isolates from the Latin American Mediterranean genotype family with the st. 33, spoligotype 776177607760771 (isolated from CNS TB patients), to invade the CNS. They show that viable Mtb can be detected in brain homogenates of BALB/c mice infected by the intranasal route, before microbiological or histologic evidence of pulmonary disease, and propose a hypothesis, that certain Mtb strains bypass the lungs and can directly colonize the CNS as early as 1-3 days after infection.
Although this hypothesis is very exciting and if it proves to be true, it will be a very significant addition of this field, since the precise mechanisms regulating CNS dissemination of Mtb is not completely understood. However, I think the authors do not show sufficient evidence to support their hypothesis. Although the CFU results might suggest true neurotropism for the studied Mtb clinical isolates, a clear conclusion cannot be drawn from the data presented. Although the mice have been exsanguinated, it cannot be proven that the CFU is originating from the tissue parenchyma and not from inside the brain vasculature or the surface of the meninges, i.e., if the BBB was indeed breached by these Mtb strains. There is also not sufficient evidence presented to prove disseminaton via the olfactory nerves and not from an early hematogenous dissemination event. The authors describe the presence of early lung histopathology (day 7-14 and onward) in mice infected with the clinical isolates, compared to the slower course of events (day 14-21) in the case of infection with H37Rv (Results, rows 211-221). This means it is possible that Mtb clinical isolates are not controlled and show hematogenous dissemination at an early time point after infection, from the lungs. This is also supported by the presence of CFU in the kidney at 3 days post infection with isolate 209 (Figure 7A) or in the spleen at 7 days post infection with isolate 136 (Figure 7C). If the authors provided histological evidence for the presence of Mtb in the CNS tissue at an early time point, it would help to prove their statement about neurotropism and route of dissemination. IHC staining of the olfactory nerves, lymphatic vessels and Mtb, or other methods that provide spatial information and presence of Mtb would also help to prove their proposed alternative route of CNS infection. In a referenced previous study, the authors show the presence of Mtb in the CNS but only at a later time point, at which the route of CNS dissemination also cannot be established.
The authors show that CFU of isolates 209 and 136 appears in the olfactory bulb and in the brain stem at the same, very early time point: at day 7. If the route of infection is through the olfactory nerves and olfactory bulb and it is not hematogenous, wouldn't we expect to see the infection at a the brain stem later than in the OB?
Technical questions:
It is not clear how many mice were used for the survival experiments. The authors mention 65 mice per group (row 157), and state experiments were repeated 3 times (row 157), then it is stated that 20 mice per group was used to record survival (row 158). In the legend of Figure 1, the use of 20 mice are mentioned again. How many mice were used how many times? What does "as before" mean in row 157? A citation or a supplemental data table containing the results of the independent experiments would be helpful for the reader.
In section 2.5 Tor the determination of CFU in tissue homogenates, the authors mention two methods: first, the viable count determination method (previously described in section 2.2 as fluorescent staining of live and dead bacteria and measurement by flow cytometry) and second, plating the tissue homogenate on 7H10 plates and counting CFU. Which method was used to gain the CFU numbers shown on Figures 1, 4 and 7? If the authors used flow cytometry, it would be helpful to see an example for the gating strategy (especially in the case of tissue homogenates where substantial amount of cell debris is present), as well as an added citation for this staining method for Mtb (row 134).
Figures 1, 4 and 7: Several data points do not have error bars. In these cases, did the 3 mice have identical results or were some values excluded/animals died?
Figures 1, 4 and 7: What is the limit of detection for the CFU measurements in the different organs? In multiple cases, the H37Rv (control) group seems to fall below the limit of detection (all values in that group are zero or not visible on the graph because of the y axis scaling). Comparison of a group that is below the LOD to another group (which has non-zero mean and SD values) with Student's T-test might not be appropriate, since the variances of the compared groups are surely different. Increasing the number of animals per group or using a more sensitive Mtb detection method might help with this issue.
Figures 5. and 6. show the lack of infection and immunopathology in the CNS and the olfactory mucosa at 120 days post infection (DPI) in the Mtb strain 209-infected group. This corroborates the results shown in Figure 1C, which suggested clearance of Mtb strain 209 from the CNS by 120 DPI. Since the main statement of the paper is that neurotrophic Mtb strains can reach the CNS early, it would be just as important to show histology from these anatomical regions at an earlier time point, i.e. 7-14 DPI, when the CFU of Mtb strain 209 peaks in the brain tissue homogenate.
Row 300: strain 136 CFU is not negative in the olfactory bulb at day 120, as the authors correctly state this in the second part of the sentence.
I recommend to present more evidence regarding the early events of infection that supports the proposed hypothesis before publication.
Comments on the Quality of English Language
The manuscript is easy to read but the methods section could be a little clearer (see comments above) and some typos can be found in the manuscript, e.g. "Long Rank test" vs "Log Rank test".
Author Response
REVIEWER 1
Response to the comment:
In the scope of human tuberculosis's natural history, the current validated evidence centers on respiratory transmission and its lymphohematogenous spread in disseminated and miliary forms.
Our proposed hypothesis attempts to shed light on a perplexing scenario frequently observed in patients from our institute: the emergence of progressive neurological deficits, often unaccompanied by specific febrile syndromes, and lacking evidence of imaging abnormalities, M. tuberculosis culture, or other microorganisms. Clinical assessments including ADA or PCR studies, CSF culture, or Ziehl-Neelsen staining have failed to pinpoint any infectious focus. Only in some post-mortem cases has the presence of neurotuberculosis been confirmed, unveiling tuberculomas, vasculitis, or abscesses without systemic or pulmonary connections. We hypothesize that in select individuals, an alternate dissemination pathway might exist, likely influenced by nasal mucosa damage or changes in the immune response of the mucosa-associated lymphoid tissue (NALT), coupled with distinctive "immunotropism" traits found within specific strains.
Remarkably, intranasal infection with the reference strain H37Rv leads to lung infection, albeit at a later stage than brain infection. Intriguingly, this strain does not invade the olfactory bulb, indicating that H37Rv triggers brain infection through hematogenous dissemination in the advanced stages of active pulmonary disease.
Hernandez-Pando et al.'s study (2010) revealed the H37Rv strain's strong lung affinity, causing immediate infection upon direct intratracheal inoculation. Bacterial virulence significantly influences its bloodstream dissemination to other organs, maintaining concentrations below one million CFUs. However, brain infection typically arises in the disease's later stages, likely due to tissue damage (pneumonia) and high pulmonary bacillary burdens leading to bloodstream dissemination.
Conversely, infecting lungs with strain 209 via the intratracheal route resulted in swifter pneumonia and death, indicating higher virulence. From day 7 post-infection, it displayed faster bloodstream dissemination to the brain, peaking at 28 days. This was evidenced by meningeal and perivascular inflammation, alongside bacilli presence. Strains 28 and 136 followed a more insidious course, infecting the lungs, blood, spleen, and liver more rapidly, albeit prolonging mouse survival.
These findings lead us to speculate that these clinical strains may exhibit a predilection for the central nervous system (CNS) compared to the reference strain H37Rv. In our present study, employing the intranasal route with similar concentrations, we noted the H37Rv strain's lung affinity but lacking the same virulence observed with intratracheal inoculation. Isolation and growth of the H37Rv strain from the brain were not observed until day 120, with CFU counts remaining below 1x104. In contrast, strains 209, 136, and 28 infect the brain early but with limited CFU counts. These bacteria might reach the brain via hematogenous routes from the lungs or other infected organs, as described. Alternatively, the olfactory nerve could serve as a pathway for these bacteria, akin to leprosy and certain Herpesviridae family viruses, as discussed.
The minimal bacterial load in the brain explains the absence of inflammatory infiltrates or microglial nodules. However, notable interstitial edema and neuronal morphological alterations, akin to Hernandez-Pando et al.'s 2010 study, were observed.
The disparity between both experiments suggests that a higher bacterial load of M. tuberculosis tends to follow the respiratory route, establishing itself in the lungs and then disseminating. However, certain strains, like H37Rv, might selectively remain in the lungs, while others tend to establish in the brain under favorable conditions. This raises questions about the brain possibly serving as a latent state niche, leading to cerebral infection, regardless of a systemic origin, or possibly through other routes like neurotropism.
Currently, as part of this research, we're developing a murine model with shorter intranasal infection times. Using ZN and indirect immunohistochemistry methods, we aim to trace the microorganism, observing changes via histopathology, etc. However, our results are inconclusive at this stage.
In conclusion, this work presents an interesting proposal, urging input from international experts and critics to elucidate this possible mechanism and further our research. Acknowledging our partial results, we seek to explore this proposed dissemination via nasal cavity structures. The clinical manifestation of brain infection without evident demonstration in other organs merits investigation to establish or discard its possible pathophysiology in the future.
Technical questions
It is not clear how many mice were used for the survival experiments. The authors mention 65 mice per group (row 157), and state experiments were repeated 3 times (row 157), then it is stated that 20 mice per group was used to record survival (row 158). In the legend of Figure 1, the use of 20 mice are mentioned again. How many mice were used how many times? What does "as before" mean in row 157? A citation or a supplemental data table containing the results of the independent experiments would be helpful for the reader.
The number 65 was a typo. Since there were three independent experiments with 20 mice per strain, arithmetic would give us 60 animals per group. Indeed, this, along with the wording, complicated the understanding of the paragraph, which has now been corrected in lines 156-159:
Three whole heads were reserved to evaluate encephalon and nasal cavity structures in a full morphofunctional context. A survival study was performed for 16 weeks. 20 mice from each group were left undisturbed to the record. Three separate experiments were performed.
In section 2.5 Tor the determination of CFU in tissue homogenates, the authors mention two methods: first, the viable count determination method (previously described in section 2.2 as fluorescent staining of live and dead bacteria and measurement by flow cytometry) and second, plating the tissue homogenate on 7H10 plates and counting CFU. Which method was used to gain the CFU numbers shown on Figures 1, 4 and 7?
From line 105 to 142, the methodology is described for the bacterial inoculum used to infect the mice. Implicitly, in line 108, reference is made to the inoculation of a liquid medium with an axenic culture preserved at -80ºC, without recovering bacteria from any tissue. Since a direct viable count was required for inoculation at the time, flow cytometry was used to adjust the inoculum, which was diluted and seeded onto 7H10 plates to confirm the estimated concentration, allowing for a variation of plus/minus 2.5% of CFUs compared to the concentration calculated with the cytometer.
Specifically, it has been corrected in line 108:"Briefly, -80ºC frozen aliquots of mycobacterial suspension was thawed, 0.5 mL of these used to…
From line 171 to 185, a homogenizer is used for the tissues to determine the bacterial load. The size of the glass beads used in the homogenizer breaks eukaryotic cells, releasing bacteria from within and between cells. From the tissue lysates, viable counts can be performed using dilutions.
If the authors used flow cytometry, it would be helpful to see an example for the gating strategy (especially in the case of tissue homogenates where substantial amount of cell debris is present), as well as an added citation for this staining method for Mtb (row 134).
In this case, the technique was used to calculate the bacterial load from axenic cultures, so there were no eukaryotic cell remnants present that could cause interference in the inoculum quantification. Handle of infected tissue homogenates is quiet complicated because biosecurity matters.
Figures 1, 4 and 7: Several data points do not have error bars. In these cases, did the 3 mice have identical results or were some values excluded/animals died?
Response: In figures 1, 4, and 7, the absence of bars for certain groups indicates identical values (where the standard deviation is zero).
Figures 1, 4 and 7: What is the limit of detection for the CFU measurements in the different organs?
To our knowledge, there isn't a microbiological test for validating the development of CFUs (Colony Forming Units) in tissues. In theory, bacterial growth on a solid culture medium allows for CFU development. However, for bacteria like Mycobacterium tuberculosis (MTb), bacteriological factors need consideration. The latent state of MTb can render them viable but not culturable, and these might not be directly implicated in tissue damage. Searching for mycobacteria via cytometry in cell lysates might face interference from cellular debris in the mouse tissues. The only other type of test we consider that could indirectly validate bacterial load would be qPCR associated with mRNA reverse transcription of a constitutive gene and comparison with a standard curve.
In multiple cases, the H37Rv (control) group seems to fall below the limit of detection (all values in that group are zero or not visible on the graph because of the y axis scaling). Comparison of a group that is below the LOD to another group (which has non-zero mean and SD values) with Student's T-test might not be appropriate, since the variances of the compared groups are surely different.
Upon reviewing and reconsidering this study, we concur that indeed the appropriate analysis was a two-way ANOVA, which has been conducted and included in the main body of the work, resulting in changes in statistical significance.
Increasing the number of animals per group or using a more sensitive Mtb detection method might help with this issue.
With regards to the Mycobacteria detection technique, as far as our knowledge goes, there isn't a more efficient direct counting technique than viable counts. As mentioned earlier, indirect or mechanized techniques have their disadvantages. In comparison with other studies, e.g., van Well 2007, our model has a higher statistical power due to the number of animals. Moreover, with the modifications in animal use regulations in our country, employing such large groups of animals is no longer feasible.
Figures 5. and 6. show the lack of infection and immunopathology in the CNS and the olfactory mucosa at 120 days post infection (DPI) in the Mtb strain 209-infected group. This corroborates the results shown in Figure 1C, which suggested clearance of Mtb strain 209 from the CNS by 120 DPI. Since the main statement of the paper is that neurotrophic Mtb strains can reach the CNS early, it would be just as important to show histology from these anatomical regions at an earlier time point, i.e. 7-14 DPI, when the CFU of Mtb strain 209 peaks in the brain tissue homogenate.
At the moment, we are conducting an assay testing shorter times and carrying out a more detailed analysis of nasal cavity histology and associated structures. However, in this case, we consider the standardization of the intranasal infection model relevant, which is not described for the development of pulmonary TB. Based on the results showing a higher bacterial load at the olfactory bulb, we propose a new infection route and are already working on a more detailed assay.
Row 300: strain 136 CFU is not negative in the olfactory bulb at day 120, as the authors correctly state this in the second part of the sentence.
Indeed, there is growth of strain 136 observed on day 120 in the olfactory bulb but not in the brainstem. And the correction has been made Line 258-264
Reviewer 2 Report
Comments and Suggestions for Authors
The authors have improved the manuscript. However, there still lies minor grammatical errors and spelling mistakes (for example, parenchyma has been written as parenchima) in the manuscript.
Comments on the Quality of English Language
The grammatical errors and spelling mistakes should be corrected.
Author Response
REVIEWER 2
Minor grammatical errors and spelling mistakes corrected.
Reviewer 3 Report
Comments and Suggestions for Authors
This study still maintains the theme where 3 different clinical strains were tested and intranasally-administrated in the BALB/c animal model. One of the strains had a very fast rate of reaching the olfactory bulb and therefore the nasal cavity was studied in more detail with indications that previous primary disease not necessary for certain strains to infect the CNS. Traditionally it is thought that CNS-TB culminates after pulmonary infection and thereafter spreads to different organs which seems not to necessarily be the case for some clinical strains. This has also been referred to in the introduction section to Arvanitakis Z., et al 1998 and Caws M. et al 2008 which has indicated similar conclusions.
The authors have now indicated that the difference in the approach of the study of Hernandez-Pando R et al., 2010 and the current study were that of the route of infection. Where the 2010 study was done with intratracheal infection vs intranasal infection in the current study with the observed differences noted from these two.
As the bacteria was observed in the early stages of infection in the olfactory bulb a suggestion was made that the infection might be through the peripheral nerves (probably Schwann cells) or lymphatic vessels of the nasal mucosa although no pathological changes were observed in these regions. These conclusions were drawn in relation to what is observed during the infection of Mycobacterium leprae. The authors have now underlined that more information and investigations are required in this regard and that at this stage the conclusions are hypothetical.
Line 55: disease
Line 81-81: blood-brain barrier
Line 84-86: re-write to: Furthermore, evidence suggests that a pulmonary infection is not necessarily the primary area of infection, and that certain Mtb genotypes have been described that could have a predilection to colonize and invade the CNS in the early stages of infection, coining the terms neurovirulent and neurotropic strains.
Line 197 re-write: The only significant difference (p <0.01) observed were between strain 136 and the control strain.
Line 216: followed
Line 217: kinetics
Line 219: change “spotted” to “observed”
Line 415: M. bovis BCG
Comments on the Quality of English Language
As per the above comments
Author Response
REVIEWER 3
-Line 55: disease
The error has been corrected
-Line 82-83: blood-brain barrier
The error has been corrected
-Line 85-: re-write to: Furthermore, evidence suggests that a pulmonary infection is not necessarily the primary area of infection, and that certain Mtb genotypes have been described that could have a predilection to colonize and invade the CNS in the early stages of infection, coining the terms neurovirulent and neurotropic strains.
The paragraph has been rewritten
-Line 197 re-write: The only significant difference (p <0.01) observed were between strain 136 and the control strain.
The paragraph has been rewritten Line 198
-Line 216: followed
The paragraph has been rewritten Lines 215-217
-Line 217: kinetics
The paragraph has been rewritten Lines 215-217
-Line 219: change “spotted” to “observed”
The error has been corrected Line 220
-Line 415: M. bovis BCG
The error has been corrected Line 374
Reviewer 4 Report
Comments and Suggestions for Authors
The article entitled "Central nervous system tuberculosis in a murine model: Neurotropic strains or a new pathway of infection?" by Rembao-Bojórquez et al described the possibility of Mycobacterial infection in CNS by using different Mtb strains. It is exciting to see how different Mtb strains appear in the tissues at different time points, which indicates a possibility of diverse infection pathways by mycobacteria. This makes the article interesting and has the potential to show different infection routes for Mtb infection in mice that can also apply to tuberculosis disease in humans.
However, the article needs further correction and modification for better clarity of the results. Below are my comments:
(1)In the result section, instead of putting keywords such as “survival”, “lung histology,” etc., in the subheading of the result section, please write the observation or message from that particular figure. Writing only keywords does not make any sense or provide the readers with a clear idea about the result sections.
(2)Figure-1B: Authors mentioned that, all the Mtb strain’s CFU count in lung were significantly low compared to the H37Rv which is only valid for specific time points. On day 14 and day 21 post-infection, the results are different. Please mention the time points while discussing this data.
(3)Authors have included the CFU encephalon in figure1 and discussed it with figure4. It is creating confusion. It will be helpful for the readers to combine the figures as they appear in the result section.
(4) Figure 2: Mention the scale bar in the histology images. Also, for Figure 2 B and C, Mark the area with the arrow the authors refer to.
How is the acid-fast stain for the same bacteria strain at these time points?
(5)Figure caption of Figure 3: Mention what AFB is when using the abbreviation for the first time in the text.
(6)Figure 3: Similar to the previous comment as figure 2: please mention scale bar here as well.
(7) Line 215-216: “Whereas mice infected with the two other clinical isolates showed similar histopathological alterations and follow the same kinetic.”
Same kinetic with what? (Please mention strain name clearly)
(8)It is impossible to see the the CFU count on day 1 post infection on the graphs in figure 1. Please separate the CFU graphs for day1-3 and day 7-120 for clear vizualization.
(9)Figure 5 and 6: Authors mentioned that there is no acid fast positivity in the histology for the strain 209 at day 120 post infection. From the CFU graphs it is clear that at day 120 the tissue brae negative for strain 209. It is expected that histology may also come negative for the sign of Mtb. It is more helpful if the authors can show the acid-fast stain at the time points where there is sign of Mtb according to the CFU data. This way, it will be easier to compare with the lung histology data.
(10)Figure6: What is the inflammation status of the olfactory area between day 7 to day 14 post-infection when the bacteria load of strain 209 is at its peak? Is it possible that, the tissue undergoes inflammatory and infection-related changes between day7-14 and then recover to normal stage towards day 120?
(11) Typing error: Figure 7ba . Isn’t it Figure 7b ? (Line number: 333)
(12) Figure 7: For the CFU graphs, please remake the graphs in such a way that the reader can see the values of H37Rv. The values of H37Rv are on the X-axis, and it is very difficult to see how the data are compared between groups.
Author Response
REVIEWER 4
(1)In the result section, instead of putting keywords such as “survival”, “lung histology,” etc., in the subheading of the result section, please write the observation or message from that particular figure. Writing only keywords does not make any sense or provide the readers with a clear idea about the result sections.
Changed to:
Superior lethality, pathogenicity and pulmonary dissemination of 209 strain. Line 193
2)Figure-1B: Authors mentioned that, all the Mtb strain’s CFU count in lung were significantly low compared to the H37Rv which is only valid for specific time points. On day 14 and day 21 post-infection, the results are different. Please mention the time points while discussing this data.
Indeed, the reviewer observation is accurate, and it has already been corrected.
In the evaluation of pulmonary bacterial load, we found that strains exhibited significantly lower infection to the lung, as demonstrated by viable CFU counts compared to the control strain H37Rv throughout almost all the evaluated time points except for days 14 and 120, where strain 136 presented a higher load.
(3)Authors have included the CFU encephalon in figure1 and discussed it with figure4. It is creating confusion. It will be helpful for the readers to combine the figures as they appear in the result section.
We agree with the reviewer and have modified the image as suggested.
(4) Figure 2: Mention the scale bar in the histology images. Also, for Figure 2 B and C, Mark the area with the arrow the authors refer to.
How is the acid-fast stain for the same bacteria strain at these time points?
The scale bars and markers have been added to the image.
(5)Figure caption of Figure 3: Mention what AFB is when using the abbreviation for the first time in the text.
The observation is accurate, and it has already been corrected.
(6)Figure 3: Similar to the previous comment as figure 2: please mention scale bar here as well.
The scale bars and markers have been added to the image.
(7) Line 215-216: “Whereas mice infected with the two other clinical isolates showed similar histopathological alterations and follow the same kinetic.”
Same kinetic with what? (Please mention strain name clearly)
Whereas mice infected with the strains 28 or 136 showed similar histopathological alterations at the same time points as the observed in mice infected with 209 strain. Line 237-239
(8)It is impossible to see the CFU count on day 1 post infection on the graphs in figure 1. Please separate the CFU graphs for day1-3 and day 7-120 for clear visualization.
The scale of the Figure 1 was adjusted to observe the CFUs in the early days.
(9)Figure 5 and 6: Authors mentioned that there is no acid fast positivity in the histology for the strain 209 at day 120 post infection. From the CFU graphs it is clear that at day 120 the tissue brae negative for strain 209. It is expected that histology may also come negative for the sign of Mtb. It is more helpful if the authors can show the acid-fast stain at the time points where there is sign of Mtb according to the CFU data. This way, it will be easier to compare with the lung histology data.
Although the microbial load was determined by CFUs and the presence of the bacteria confirmed through PCR, multiple different cutoff levels were tested without identifying changes that could pinpoint the focus of infection or guide us towards locating the mycobacteria. Additionally, with ZN staining, we were unable to detect any positive bacteria. It is partly for these reasons that it is postulated that the bacteria might be in a physiological state of latency, where carbohydrate metabolism is altered, leading to a loss of acid-fast bacilli positivity.
(10)Figure6: What is the inflammation status of the olfactory area between day 7 to day 14 post-infection when the bacteria load of strain 209 is at its peak? Is it possible that, the tissue undergoes inflammatory and infection-related changes between day7-14 and then recover to normal stage towards day 120?
That's possible. In fact, we're currently conducting experiments with earlier evaluation times to precisely determine changes in nasal and craniofacial structures and gain a better understanding of the Mycobacteria entry process.
(11) Typing error: Figure 7ba . Isn’t it Figure 7b ? (Line number: 333)
Indeed, the observation is accurate, and it has already been corrected. Line 297
(12) Figure 7: For the CFU graphs, please remake the graphs in such a way that the reader can see the values of H37Rv. The values of H37Rv are on the X-axis, and it is very difficult to see how the data are compared between groups.
The scale of the Figure 7 was adjusted to observe the CFUs of H37Rv.
Round 2
Reviewer 1 Report
Comments and Suggestions for Authors
I thank the authors for their clear explanation. Their response answered my questions and I agree with their point, that even if the exact mechanism of CNS entry is not understood yet, their discovery about the early CNS dissemination of certain strains, and the possibility that latent infection can exist in the brain is thought provoking and would serve as a valuable basis for further research.